# Automatic Detection and Measurement of Renal Cysts in Ultrasound Images: A Deep Learning Approach

**DOI:** 10.3390/healthcare11040484

**Published:** 2023-02-07

**Authors:** Yurie Kanauchi, Masahiro Hashimoto, Naoki Toda, Saori Okamoto, Hasnine Haque, Masahiro Jinzaki, Yasubumi Sakakibara

**Affiliations:** 1Department of Biosciences and Informatics, Keio University, Yokohama 2238522, Japan; 2Department of Radiology, Keio University School of Medicine, Tokyo 1608582, Japan; 3GE HealthCare Japan, Tokyo 1918503, Japan

**Keywords:** deep learning, ultrasonic imaging, kidney, object detection

## Abstract

Ultrasonography is widely used for diagnosis of diseases in internal organs because it is nonradioactive, noninvasive, real-time, and inexpensive. In ultrasonography, a set of measurement markers is placed at two points to measure organs and tumors, then the position and size of the target finding are measured on this basis. Among the measurement targets of abdominal ultrasonography, renal cysts occur in 20–50% of the population regardless of age. Therefore, the frequency of measurement of renal cysts in ultrasound images is high, and the effect of automating measurement would be high as well. The aim of this study was to develop a deep learning model that can automatically detect renal cysts in ultrasound images and predict the appropriate position of a pair of salient anatomical landmarks to measure their size. The deep learning model adopted fine-tuned YOLOv5 for detection of renal cysts and fine-tuned UNet++ for prediction of saliency maps, representing the position of salient landmarks. Ultrasound images were input to YOLOv5, and images cropped inside the bounding box and detected from the input image by YOLOv5 were input to UNet++. For comparison with human performance, three sonographers manually placed salient landmarks on 100 unseen items of the test data. These salient landmark positions annotated by a board-certified radiologist were used as the ground truth. We then evaluated and compared the accuracy of the sonographers and the deep learning model. Their performances were evaluated using precision–recall metrics and the measurement error. The evaluation results show that the precision and recall of our deep learning model for detection of renal cysts are comparable to standard radiologists; the positions of the salient landmarks were predicted with an accuracy close to that of the radiologists, and in a shorter time.

## 1. Introduction

Ultrasonography is widely used for diagnosis of diseases in internal organs, such as the abdomen, heart, and thyroid gland, as well as for prenatal diagnosis. This is because it is nonradioactive, noninvasive, real-time, and inexpensive. In ultrasonography, a set of measurement markers is placed at two points to measure organs and tumors, and the position and size of the target finding are measured based on these. While the markers play an important role in diagnosis, they must be manually positioned, which places a burden on the sonographer. Furthermore, because of the absence of fixed rules on the placement of markers, differences between individuals with different levels of experience create an additional problem [1]. Therefore, we believe that automating the placement of markers using deep learning would lead to a reduced burden on sonographers, shorten the test duration, and promote the elimination of inter-operator variability.

Studies have been conducted to automate measurement in ultrasonography using deep learning. In ultrasonic image analysis, deep learning systems for detection and classification of thyroid nodules, breast lesions, and liver lesions have been developed [2]. Ma et al. [3] developed a system to automatically detect thyroid nodules from ultrasound B-mode images using a cascade model containing two CNN models, achieving an area under the curve (AUC) of 98.51%. Li et al. [4] employed a CNN model consisting of ResNet 50 and Darknet pre-trained with ImageNet [5] for the diagnosis of thyroid cancer. Using large datasets collected from three hospitals, they showed that diagnosis with the same sensitivity and higher specificity as that of radiologists is possible. Byra et al. [6] applied transfer learning using VGG29 pre-trained with ImageNet [5] to a breast ultrasound image dataset in order to classify benign and malignant breast cancer lesions. They achieved an AUC of 93.6% on their own dataset and an AUC of approximately 89.0% on public datasets. Liu et al. [7] proposed a system that extracts liver membrane features from ultrasound images using a pre-trained CNN model, then classifies liver normality and abnormalities by support-vector machine (SVM) based on the extracted features. These deep learning methods are expected to help radiologists to shorten interpretation times and improve the accuracy of diagnosis.

In addition to ultrasonic images, deep learning is widely applied to radiographic images such as X-ray and Computed Tomography (CT) images. Muresan et al. [8] proposed an approach for automatic tooth detection and disease classification in panoramic X-ray images using deep learning-based image processing. Semantic segmentation and object detection were used to detect tooth regions and label diseases affecting the tooth. Pang et al. [9] proposed a two-step segmentation framework called SpineParseNet for child spine analysis in a volumetric MR image. This model consists of a 3D graph convolutional segmentation network (GCSN) that performs 3D segmentation and a 2D residual U-Net (ResUNet) that performs 2D segmentation. Gu et al. [10] proposed an attention-based CNN (CA-Net) for more accurate and explainable medical image segmentation. Their CA-Net achieved higher accuracy than U-Net in skin lesions, placenta, and fetal brain segmentation. Takeuchi et al. [11] constructed a system for diagnosing esophageal cancer from CT images based on deep learning, and verified its performance. VGG16, which is one of the CNN models used for image recognition, was fine-tuned for classification of the presence or absence of esophageal cancer, and the authors confirmed that esophageal cancer could be detected with high accuracy. Other studies have attempted the detection of landmarks in radiographic images. Payer et al. [12] proposed a system using CNN to predict the position of a landmark in an X-ray image of the hand using heat map regression. Zhong et al. [13] applied landmark regression using a two-stage U-Net to detect anatomical landmarks on cephalometric X-ray images.

Several previous studies have been conducted to automate measurements in ultrasonography using deep learning. Chen et al. [14] proposed a method for the automatic measurement of the width of the fetal lateral ventricle. The lateral ventricle was segmented using a deep convolutional network, and measurement was performed by finding the minimum bounding rectangle of the segmented region. Biswas et al. [15] proposed an automatic measurement method for carotid intima–media thickness. The arterial wall is composed of the intima, media, and adventitia, and the intima–media thickness is the combined thickness of the intima and media. The luminal and media/adventitia regions were segmented using a deep learning method, and the intima–media thickness was obtained from the distance between the boundary surfaces of the regions. Leclerc et al. [16] conducted a study to automatically measure left ventricular volume from echocardiographic images by segmentation using deep learning methods. In these methods, segmentation is performed for measurement; however, creating a mask that serves as training data for segmentation is a time-consuming task. Although research is being conducted to automate measurements in ultrasonography of the heart, carotid artery, and foetation, there have been few studies on abdominal ultrasonography. For example, Jagtap et al. [17] proposed a method to measure the total kidney volume from 3D ultrasound images using CNN. Akkasaligar et al. [18] developed a method for automatic segmentation of renal cysts in ultrasound images using the active contour method and level set segmentation method. However, neither study addressed the prediction of salient landmark positions. Related studies on deep learning-based analysis of ultrasound images are summarized in the Table 1.

Among the measurement targets of abdominal ultrasonography, renal cysts occur in 20–50% of the population regardless of age [19]. Therefore, the frequency of measurement of renal cysts is high, which leads us to believe that the effects of automating measurement would be high as well.

Saliency map regression is often used to predict the location of landmarks such as salient landmarks; however, previous studies have predicted only a fixed number of landmarks [13]. When assigning salient landmarks to renal cysts, two landmarks are assigned to one renal cyst; because multiple renal cysts may exist in one image, it is necessary to predict the position of a non-constant number of salient landmarks. Therefore, conventional saliency map regression, which can predict only a certain number of landmarks, cannot be used to address this problem.

The following are the main contributions of this study:We developed a measurement assistance function for ultrasonic images using deep learning with the aim of supporting the measurement of renal cysts using measurement markers.To predict the landmarks for multiple renal cysts within one image, we developed a system in which all renal cysts in the image were detected prior to saliency map regression. Then, we performed saliency map regression to predict the positions of two salient landmarks for each detected renal cyst.Because the proposed method only uses the coordinates of the measurement markers when training the models, it is possible to automate the measurement without performing segmentation, thereby avoiding high annotation costs.In comparative tests, our method achieved almost the same accuracy as a radiologist. The errors of the measured length and measurement marker coordinates were used as evaluation indices.Our results indicate that the proposed method is able to perform measurements at a higher speed than manual measurement and with an accuracy close to that of sonographers.

## 2. Materials and Methods

In this paper, we propose an automated system for detecting renal cysts from abdominal ultrasonography and assigning a pair of salient landmarks to the detected renal cysts, as shown in Figure 1. The renal cyst measurement task was divided into three steps. First, a YOLOv5 object detection model [20] was trained to detect renal cysts from ultrasound images. Next, the area around each detected renal cyst was extracted from the image, and a heat map expressing the positions of the two measurement markers within that range was predicted using the UNet++ convolutional neural network [21]. Finally, the output heat map was post-processed and the coordinates were corrected to determine the predicted coordinates of the measurement marker. First, we developed a system in which all renal cysts in the image were detected prior to saliency map regression. Then, we performed saliency map regression to predict the positions of two salient landmarks for each detected renal cyst. Thus, two models were required for one for the task of detecting renal cysts and another for saliency map regression to predict the positions of the salient landmarks. The output of the saliency map regression was subjected to post-processing in order to determine the appropriate coordinates of each salient landmark. If the size of the detected renal cyst and measurement result using the predicted salient landmark coordinates were significantly different, the predicted coordinates were corrected. We integrated a renal cyst detection model, a salient landmark position prediction model, and coordinate determination with post-processing and correction to construct an automated system for salient landmark assignment. The renal cyst detection model and the salient landmark position prediction model were trained separately. The performance of this system was compared with that of a radiologist. The system was trained and evaluated using 2664 ultrasound images of renal cysts.

### 2.1. Automated System for Assigning Salient Landmarks

Figure 2 shows the processing flow of an automated system that integrates the renal cyst detection model, the salient landmark position prediction model, and coordinate determination by post-processing and correction. First, an ultrasound image was input to the renal cyst detection model and a bounding box surrounding the renal cyst was output. The area surrounded by the output bounding box was extracted from the ultrasound image. Because this process was performed on all bounding boxes output by the renal cyst detection model, a multiple number of bounding boxes and their areas were extracted. Then, the extracted area images were input to the salient landmark position prediction model one by one and a saliency map denoting the position of the salient landmark was output. For each of the obtained saliency maps, post-processing and correction of coordinates were performed as necessary to determine the appropriate coordinates of the salient landmarks. Because salient landmark position prediction was performed for all the bounding boxes output by the renal cyst detection model and because the salient landmark position prediction model predicts the positions of two salient landmarks per bounding box, the number of output salient landmarks was twice the number of detected renal cysts.

### 2.2. Renal Cyst Detection Model

We constructed a model to detect renal cysts using ultrasound images as input. A renal cyst detection model was trained to predict the bounding box surrounding the renal cyst. The YOLOv5 object detection algorithm was used as the model. YOLOv5 is a model originally proposed by Glenn Jocher in June 2020 [20]; its architecture is shown in Figure 3. Compared with object detection models that require two steps for prediction (i.e., searching for area candidates in which an object appears from an image and identifying its category), YOLO directly predicts the bounding box and its class [22]. Therefore, the calculation speed of YOLO is higher than that of the conventional methods. Moreover, the entire image is used during training, making it possible to consider the surrounding context. Because the detection of renal cysts requires information on whether the background is the kidney, a model that can detect objects based on the surrounding context is suitable. In addition, when the automated system is used for ultrasonic examination, the processing must be faster than the manual placement of salient landmarks by the sonographer. For these reasons, YOLOv5 was selected as a suitable model for this task. The initial parameters of YOLOv5 were pre-trained using the COCO dataset [23], which is a large dataset of RGB images with object bounding boxes and category information. YOLOv5 has multiple models of different sizes, and we used the small (YOLOv5s), medium (YOLOv5m), large (YOLOv5l), and extra-large (YOLOv5x) models for accuracy comparison. The architecture of YOLOv5 consists of three components: BackBone, PANet, and Head. Here, Bottleneck CSP [24] represents the CSP bottleneck architecture proposed by CSPNet, Conv represents the convolutional layer, Upsample represents the upsampling layer, Concat represents the concatenate function, and SPP represents spatial pyramid pooling [25], which is a pooling method that can handle images of various sizes and shapes. First, in the BackBone, namely, CSP Darknet, performs feature extraction twice on multiple scales via Conv and Bottleneck CSP. Second, pooling processing is performed on feature maps with different scales using SPP. This is the backbone of CSP Darknet, which introduces the mechanism proposed by CSPNetinto the Darknet neural network framework in order to reduce the required amount of calculation while maintaining accuracy. Third, the extracted feature map is processed by Neck. PANet is used for Neck; after repeating a series of processing of BottleNeckCSP, Conv, Upsample, and Concat twice, it is processed again by BottleneckCSP. Finally, Conv is performed in Head, and the class, score, position, and size are output as detection results. The loss function of YOLOv5 is the sum of the loss functions of the bounding box regression, confidence, and classification, as indicated in the following equations [26]:
(1)$$\begin{array}{ccc} LOSS & = & L_{GIoU}+L_{conf}+L_{class} \end{array}$$
(2)$$\begin{array}{ccc} L_{GIoU} & = & \sum_{i=0}^{S^{2}}\sum_{j=0}^{B}I_{i,j}^{obj}\left[1-IoU+\frac{A^{C}-U}{A^{C}}\right] \end{array}$$
(3)$$\begin{array}{ccc} L_{conf} & = & -\sum_{i=0}^{S^{2}}\sum_{j=0}^{B}I_{i,j}^{obj}\left[{\hat{C}}_{i}^{j}\log\left(C_{i}^{j}\right)+\left(1-{\hat{C}}_{i}^{j}\right)\log\left(1-C_{i}^{j}\right)\right]-\lambda_{nobj}\sum_{i=0}^{S^{2}}\sum_{j=0}^{B}I_{i,j}^{obj}\left[{\hat{C}}_{i}^{j}\log\left(C_{i}^{j}\right)+\left(1-{\hat{C}}_{i}^{j}\right)\log\left(1-C_{i}^{j}\right)\right] \end{array}$$
(4)$$\begin{array}{ccc} L_{class} & = & -\sum_{i=0}^{S^{2}}I_{i,j}^{nobj}\sum_{c\in classes}\left[{\hat{P}}_{i}^{j}\log\left(P_{i}^{j}\left(c\right)\right)+\left(1-{\hat{P}}_{i}^{j}\left(c\right)\right)\log\left(1-P_{i}^{j}\left(c\right)\right)\right] \end{array}$$
where $S^{2}$ is the number of grids, *B* is the number of bounding boxes in each grid, obj means that an object exists in a bounding box, nobj means that no object exists in the bounding box, $I_{i,j}^{obj}$ is equal to 1 when an object exists in a bounding box and is otherwise 0, IoU is the Intersection over Union between the predicted bounding box and the real bounding box, $A^{C}$ is the smallest rectangular box that can completely contain the predicted bounding box and the real bounding box, *U* is the sum of area of the predicted bounding box and the real bounding box, ${\hat{C}}_{i}^{j}$ is the prediction confidence of the *j*th bounding box in the *i*th grid, $C_{i}^{j}$ is the true confidence of the *j*th bounding box in the *i*th grid, $\lambda_{nobj}$ is the confidence weight when no object exists in the bounding box, ${\hat{P}}_{i}^{j}$ is the probability of predicting the detection object as category *c*, and $P_{i}^{j}\left(c\right)$ is the probability of actually being category *c*.

All layers of YOLOv5 were fine-tuned using the renal cyst detection dataset described later in this section. The same single grayscale ultrasound image was input to three channels for RGB in YOLOv5.

### 2.3. Saliency Map Representing the Location of Salient Landmarks

Our proposed method results in a saliency map predicting and indicating the position of the salient landmarks. Saliency map regression is often used to predict the location of landmarks. In this work, the heatmap is used to represent the saliency map, as shown in Figure 4. Using a Gaussian distribution centered on the position of the salient landmarks, the saliency map value gradually decreases according to the distance from the salient landmark position. Rather than using a uniform gradient, we used the Gaussian distribution to cause the loss function to converge abruptly around the salient landmark position. The saliency map value at the salient landmark position was set to 0, increasing the distance of saliency map value from the salient landmark position. The maximum value was set to 255. The radius of the Gaussian distribution was set to 50. The saliency map of the left landmark used the G channel of the RGB image format, while the saliency map of the right landmark used the R channel. All pixel values of the B channel were set to 0. The left part of Figure 4 shows the positions of the salient landmarks on the ultrasonic image (displayed as yellow crosses), while the right part of Figure 4 shows the corresponding saliency maps; the green part shows the position of the left landmark and the red part shows the position of the right landmark.

### 2.4. Salient Landmark Position Prediction Model

We constructed a salient landmark position prediction model that predicts a saliency map. For the prediction of salient landmark position, we adopted a strategy of predicting the saliency map instead of directly predicting the salient landmark position. In our study, the saliency map is represented by the heatmap. We adopted UNet++ [21] to produce a heatmap as output. UNet++ is an improved version of the U-Net deep convolutional neural network [27], which was developed for segmentation tasks. Using training data consisting of pairs of input images and heatmap outputs representing saliency maps, we trained UNet++ to produce a heatmap instead of segmentation. The salient landmark position prediction dataset (described in a later section) was used for fine-tuning of UNet++ to output a heatmap. U-Net consists of an encoder that extracts features by convolution and downsampling and a decoder that increases the resolution of the feature map by convolution and upsampling, producing segmentation results. A feature of U-Net is a skip connection that directly connects the feature map output in each layer of the encoder to the decoder. UNet++ decodes the feature map outputs at each level of the encoder and then connects them to the decoder by skip connection (Figure 5); this supplements the local features and enables more accurate area detection. Moreover, it reduces the difference in the expression of the encoder/decoder and simplifies the optimization problem [21]. The loss function was the mean square error of the saliency map. DenseNet121 [28] was used for the backbone. The initial parameters of the backbone were pretrained with ImageNet [5], which is a large dataset of RGB images.

Each node in the graph shown in Figure 5 represents a convolution block composed of a Convolution layer, a Batch Normalization layer, and a ReLu layer. This block is stacked in five layers. The down arrow indicates downsampling, the up arrow indicates upsampling, and the dashed arrow indicates the skip connection. Using the skip connection, the features from the preceding node with the same resolution are combined. In addition, upsampling combines features with different resolutions from the preceding node. This multi-scale feature aggregation is one of the improvements available with UNet++. During the fine-tuning process, all layers were trained. The loss function of UNet ++ was as follows:(5)$$\begin{array}{ccc} \mathrm{Loss} & = & \frac{1}{n}\sum_{i=1}^{n}{\left({\hat{y}}_{i}-y_{i}\right)}^{2} \end{array}$$
where *n* is the number of images, $y_{i}$ is the pixel value of the *i*th pixel of the correct image, and ${\hat{y}}_{i}$ is the pixel value of the *i*th pixel of the predicted image.

### 2.5. Post-Processing

Post-processing was performed to determine the appropriate coordinates of the salient landmarks from the salient landmark position image. In the R and G channels of the saliency map showing the position of salient landmarks output by the salient landmark prediction model, the coordinates with the smallest saliency map value were selected and the coordinates were determined as the appropriate coordinates of the two salient landmarks. Note that when multiple coordinates had the same minimum values, we followed the heuristic of selecting the one closest to the top left corner.

### 2.6. Coordinate Correction

If the size of the detected renal cyst and the distance between the two salient landmarks differed significantly, the predicted coordinates were corrected. Note that the size of the detected renal cyst is defined as the length of the shorter side of the bounding box. Multiple regions with low saliency map values may appear in one channel. If we were to simply select the coordinates in this image with the smallest saliency map value, the coordinates determined from the R channel and G channel might be too close to each other. On the the hand, in the case of a large difference between the distance of the two predicted salient landmarks and the size of the bounding box, it follows that the predicted coordinates must be incorrect. The criteria for determining incorrectness were calculated as follows. Assuming that the predicted coordinates are $p_{1}^{{}^{\prime }}$ and $p_{2}^{{}^{\prime }}$ and the length of the shorter side of the bounding box is *l*, a correction was made if the following condition (Equation 6) was satisfied: there is a difference in the Euclidean distance between the two predicted coordinates ($p_{1}^{{}^{\prime }}$ and $p_{2}^{{}^{\prime }}$) and their length (*l*) that is greater than 5% of the length (*l*). Note that the double vertical line represents the L2-norm and the single vertical line represents the L1-norm.
(6)$$\begin{array}{ccc} \left|\;|\right|p_{1}^{{}^{\prime }}-p_{2}^{{}^{\prime }}\left||-l\;\right| & > & l\times 0.05 \end{array}$$

The correction method was as follows. First, the coordinates with the lowest saliency map values in each of the R and G channels were selected, then the coordinate with the smaller value between them was selected as the position of the first salient landmark. Second, the coordinates of a point symmetrical to the first salient landmark with respect to the center of the bounding box were selected as the second salient landmark.

Figure 6 shows the coordinates before and after correction. Before correction, the distance between the two salient landmarks, indicated by the light blue crosses, was significantly different from the length of one side of the light blue bounding box; thus, the measurement was considered incorrect. After correction, the distance between the two salient landmarks, indicated by the light blue crosses, was closer to the length of one side of the bounding box. In addition, the measurement was more accurate when compared to the distance between the two true salient landmarks, indicated by the yellow crosses.

### 2.7. Evaluation Metrics

We evaluated the accuracy of renal cyst detection and salient landmark coordinate prediction by the automated system. The region of the renal cyst output by the renal cyst detection model was defined as one of following three types: true positive (TP), when the model correctly detected a renal cyst; false positive (FP), when the model identified regions that were not renal cysts as renal cysts; and false negative (FN), when the model failed to detect an existing renal cyst. To establish the criteria for correct detection of a measured renal cyst, a circle with a diameter of a straight line connecting two paired points was drawn for each of the true and predicted salient landmarks. TP was defined as a rate of the intersection over union (IoU) (defined in Equation (Equation 7)) of the two circles drawn from the true landmark coordinates and predicted ones that was greater than 0.5. The IoU threshold was determined by reference to a previous study on cyst detection [29].
(7)$$\begin{array}{ccc} \mathrm{IoU} & = & \frac{Area\;of\;Intersection}{Area\;of\;Union} \end{array}$$

Precision (Equation 8) and recall (Equation 9) were calculated as the detection accuracy.
(8)$$\begin{array}{ccc} \mathrm{Precision} & = & \frac{TP}{TP+FP} \end{array}$$
(9)$$\begin{array}{ccc} \mathrm{Recall} & = & \frac{TP}{TP+FN} \end{array}$$

The position error and diameter length error (DLE) in cyst measurements by both the AI and sonographer were used to evaluate the coordinates of the predicted salient landmarks. The position error (defined in Equation (Equation 10)) is defined as the Euclidean distance between the predicted and true coordinates. DLE (defined in Equation (Equation 11)) is defined as the absolute value of the difference between the Euclidean distance between the two predicted coordinates and the Euclidean distance between the two true coordinates. If the true coordinates are $p_{1}$,$p_{2}$ and the predicted coordinates are $p_{1}^{{}^{\prime }}$,$p_{2}^{{}^{\prime }}$, then the position error and DLE are defined as follows: (10)$$\begin{array}{ccc} \mathrm{Position}\;\mathrm{error} & = & \left|\right|p_{1}-p_{1}^{{}^{\prime }}\left|\right|+\left|\right|p_{2}-p_{2}^{{}^{\prime }}\left|\right| \end{array}$$(11)$$\begin{array}{ccc} \mathrm{DLE} & = & ||l-l^{{}^{\prime }}\left|\right| \end{array}$$(12)$$\begin{array}{ccc} l & = & \left|\right|p_{1}-p_{2}\left|\right| \end{array}$$(13)$$\begin{array}{ccc} l^{{}^{\prime }} & = & \left|\right|p_{1}^{{}^{\prime }}-p_{2}^{{}^{\prime }}\left|\right| \end{array}$$

The relationship between the loss function of each model and these evaluation indices is as follows. The loss function of YOLOv5 is the sum of three terms, namely, the loss functions of the bounding box regression, prediction confidence, and classification [26]. The loss function of the bounding box regression is calculated from the IoU of the correct and predicted bounding boxes. Because the salient landmark coordinates are predicted within the bounding box, a smaller loss function of the bounding box regression and more accurate prediction of the bounding box makes for an improvement in the position error and detection accuracy calculated from the bounding box. The loss function of UNet++ is the mean square error. When the output saliency map is closer to the correct answer, the mean square error and position error determined from the output saliency map decrease.

### 2.8. Performance Comparison of Model and Sonographers

The ground truth coordinates in the test data were set using the annotations of two board-certified radiologists (Sonographers 1 and 4). The annotations of the remaining two radiologists (Sonographers 2 and 3) and the predicted coordinates of the model were compared with the correct coordinates in order to calculate the recall, precision, position error, and diameter length error.

### 2.9. Post Hoc Evaluation by a Radiologist

In the form of a post hoc evaluation, the predictions of the automated system were manually evaluated by the most experienced radiologist (Sonographer 1). The predictions of salient landmarks by the automated system and predictions of salient landmarks placed by Sonographers 2 and 3 were presented to Sonographer 1 in an anonymous and random order. Sonographer 1 examined the ultrasound image and both predictions, pointed out FPs and FNs, and corrected the coordinates as necessary. The number of FPs and FNs, images with corrected coordinates, salient landmarks, and magnitude of the correction of coordinates were calculated and compared between the automated system and the two radiologists.

### 2.10. Deep Learning Framework and Computation Time

Python was used as the programming language. In YOLOv5, Torch 1.7.1 version or earlier was used as the framework. The number of training epochs was set to 100, and the weight at the epoch with high mAP (mean Average Precision) was used. SGD was used as the optimizer for training. The batch size was set to 16 and the size of one side of the input image was 256 pixels. In UNet++, TensorFlow GPU version 1.4.0 was used as the framework. The number of training epochs was set to 20, and the weight at the epoch when the loss function was improved was saved. The Adam optimizer was used for training. The hyperparameters, as presented in Table 2, were searched. As a result of this search, the hyperparameters indicated in bold were adopted.

The computational time required to read the ultrasound image, predict the coordinates of the salient landmarks, plot them, and display the image was measured. This experiment was performed on a computer with an Intel (R) Xeon (R)W-3235 CPU@3.30GH and an NVIDIA Quadro RTX 8000. In addition, the time required for manual assignment of salient landmarks by the radiology specialists was measured for comparison. The methodology source code is available in a public GitHub repository with the following address: https://github.com/henyo245/RenalCystMeasurement, accessed on 6 February 2023.

### 2.11. Materials

We extracted 170,538 images from 6420 abdominal ultrasound examinations taken by LOGIQE9 or LOGIQS8 GE ultrasonic devices from January 2019 to May 2020 at Keio University Hospital. Of the 6420 examinations, 2134 were identified as “renal cyst” in the report. Among the extracted ultrasound images, 2664 images were selected in which the body marker was located on the kidney and the radiologist determined that renal cysts were measured. Images with no salient landmarks or those determined by the radiologist to not be renal cysts were excluded. The 2664 images were taken from 1444 patients, and therefore contained multiple images of the same patients. All data were annotated by eight radiology technicians and ten radiologists at the clinical site. No double-checking was performed. All data were divided into training and test data at a ratio of 7:3. There was no patient overlap between the training and the test data. Training image data were preprocessed to suit the respective tasks of object detection and salient landmark position prediction. In addition, 100 images were randomly extracted from the test data and used as a dataset in order to compare the performance of the sonographers and the automated system. For the training data, the markers measured at the clinical site were directly used as the true ones. For the test data, the ground truth was regenerated by Sonographers 1 and 4. Sonographer 1 placed the landmarks, Sonographer 4 reviewed the results, and for images with differing opinions, Sonographers 1 and 4 discussed and reassigned the landmarks as ground truth. Patient informed consent for the retrospective datasets was obtained only for this current research work, and has not been confirmed for sharing outside Keio University Hospital.

We created a dataset for training of the renal cyst detection model. The ultrasound images were used as the input data. A bounding box, which is required for object detection training, was created by the following procedure. First, a circle was drawn with its diameter being a straight line connecting the two paired salient landmarks. Next, a square circumscribing this circle was drawn, which was used as a bounding box surrounding the renal cyst. The *x* and *y* coordinates, height *h*, and width *w* of the center coordinates of the bounding box were saved in a text file and used as the training data.

We created a dataset for use in training the salient landmark position prediction model. The area surrounded by the bounding box for use as the training data of the renal cyst detection model was cropped from the ultrasound image and used as the input image of the salient landmark position prediction model. Then, to ensure that the outline of the renal cyst fit in the image, the cropped area of the bounding box was resized by expanding the area by the length of one side of the bounding box × 0.2 on the top, bottom, left, and right. The cropped images were resized to 256 × 256 pixels and used as the input images. In addition, a saliency map (heatmap) representing the salient landmark position in the area corresponding to the input image was generated and used as training data.

### 2.12. Ethics

This study was approved by the Ethics Committee of the Keio University School of Medicine (ethical approval code 20170018).

## 3. Result

### 3.1. Performance Comparison of Model and Sonographers

Table 3 shows the performance of the automated system on the test data and a performance comparison on the 100 selected images between the automated system and the two radiologists (Sonographers 2 and 3) with the annotation of two specialists (Sonographer 1 and 4) as the ground truth. As a result of the automated system for assigning salient landmarks, the detection accuracy was the highest when YOLOv5m and UNet++ were used and the coordinates were corrected. The precision and recall of the automated system were comparable to Sonographers 2 and 3. The position error and the diameter length error (DLE) of the automated system were comparable or slightly lower than those of the sonographers. No significant differences were found among the YOLO models. Figure 7 shows several examples of annotations by radiologists and the coordinates predicted by the automated system on the ultrasound images.

Figure 8 shows the coordinates of the salient landmarks placed by Sonographer 1 and the coordinates predicted by the automation system.

### 3.2. Post Hoc Evaluation by Radiologist

Table 4 presents the results of the post hoc manual evaluation by the most experienced radiologist. The numbers of FPs and FNs produced by the automated system were less than those produced by Sonographer 2 and the same as those produced by Sonographer 3. The number of corrected images and salient landmarks were larger than those produced by Sonographers 2 and 3. The difference between the number of images and the number of salient landmarks is because there were ultrasound images with multiple renal cysts and images in which the coordinates of only one of the pair of salient landmarks were corrected. The magnitude of this modification was the smallest when using the proposed system.

### 3.3. Computational Time

The average execution time of the model was 0.45 ± 0.02 s per image. The average time required for measurement by a radiologist was 14.9 ± 4.5 s per image.

## 4. Discussion

There are a number of limitations to this study. First, this study was a single-center study limited to Japanese patients. In addition, the ultrasound images were taken with a limited number of ultrasonic device models. More ultrasound imaging datasets of renal cysts from multiple institutions are required to build a more accurate system. Moreover, the results may change because of the influence of the amount of noise depending on the sonographer’s imaging skills, ghost artifacts (unwanted reflections of ultrasonic waves), and artifacts, such as shadows that darken the back of tissues.

Based on the above, the salient landmark prediction system constructed using deep learning technology has great potential to detect renal cysts faster than radiologists and with comparable accuracy. Further training using larger amounts of data collected from multiple institutions can enable even more accurate detection and measurement of renal cysts. In addition, because the constructed method can be applied to other targets, such as hepatic cysts, we expect artificial intelligence-based measurement support systems for various areas of interest to be developed in the future.

There are a number of issues with the system developed in this study that could represent possibilities for future improvement. Several parameters, including the radius of the Gaussian distribution and the threshold value used for coordinate correction, were determined experimentally; in future work, a more comprehensive and systematic optimization of these parameters is necessary. We simply input the same single grayscale image into three channels for RGB in YOLOv5. Optimizing YOLOv5 to work with only one grayscale channel is another possibility for future work.

## 5. Conclusions

In this study, we constructed an automated system for assigning salient landmarks to renal cysts using deep learning methods, namely, YOLOv5 and UNet++, with 2664 ultrasound images. Previous studies have relied on segmentation, and have not targeted the abdomen. Here, we developed an automatic measurement method for renal cysts that does not require segmentation. Because the position of the salient landmarks can be predicted with an accuracy close to that of a radiologist in a shorter time, this system is useful for automating the measurement process in ultrasonography.

## Figures and Tables

**Figure 1 healthcare-11-00484-f001:**
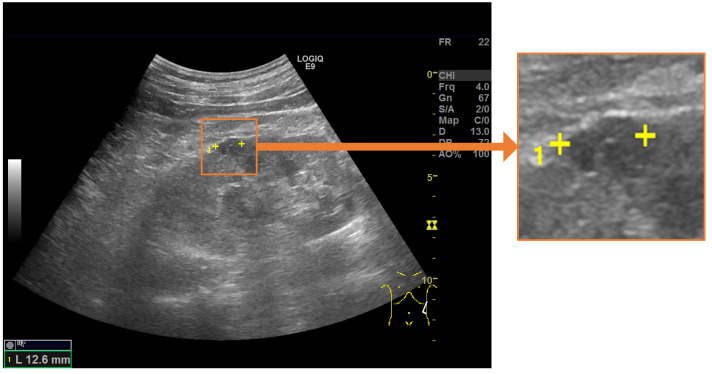
Salient landmarks placed on a renal cyst. A pair of salient landmarks are placed on the longest diameter of the renal cyst.

**Figure 2 healthcare-11-00484-f002:**
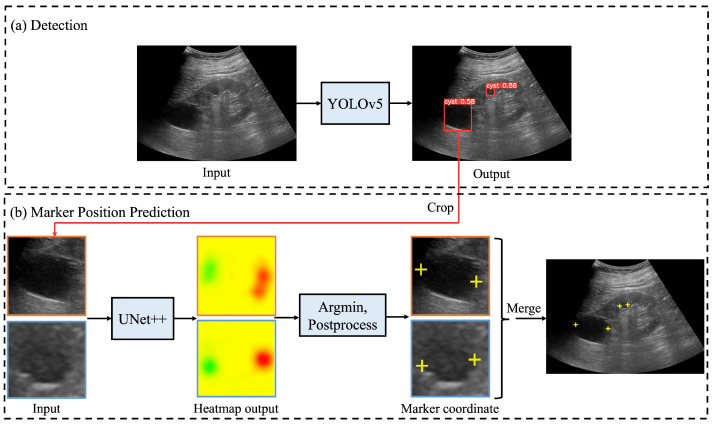
Processing flow of automated system for assigning salient landmarks to renal cysts: (**a**) ultrasound images are input into YOLOv5 [20] to detect renal cysts; (**b**) the area around the detected renal cyst is extracted and input to UNet++ [21] to predict the saliency map (represented by a heatmap) of the salient landmark position; in post-processing, the point with the smallest saliency map value from the output saliency map is selected and used as the appropriate coordinate. If necessary, the coordinates are corrected and the appropriate coordinates determined. Finally, all predicted salient landmarks are plotted on the original image.

**Figure 3 healthcare-11-00484-f003:**
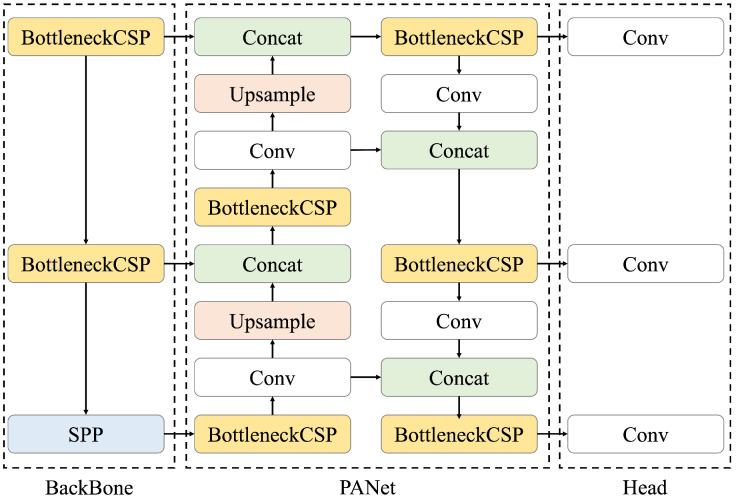
Architecture of YOLOv5, consisting of three components: BackBone, PANet, and Head. BottleNeck CSP refers to CSP Bottleneck, SPP to Spatial Pyramid Pooling, Conv to the Convolutional Layer, and Concat to the Concatenate Function. First, feature extraction is performed on multiple scales using BackBone, then PANet processes the extracted feature map, and finally Head outputs the class, score, position, and size as detection results.

**Figure 4 healthcare-11-00484-f004:**
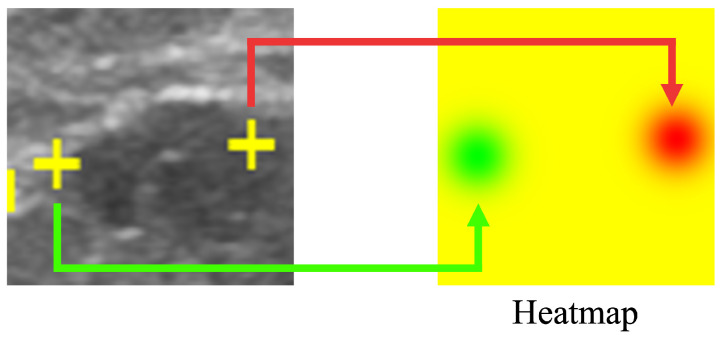
Saliency map showing the location of salient landmarks. The position of the left salient landmark is represented in green and the right one in red.

**Figure 5 healthcare-11-00484-f005:**
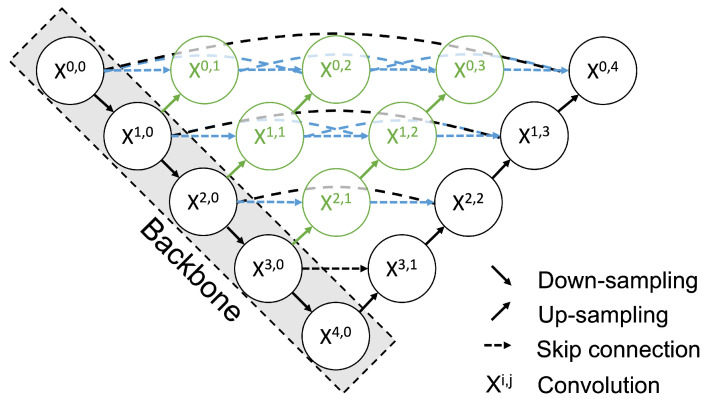
Architecture of UNet++. Each node in the graph represents a convolution block. The down arrow indicates downsampling, the up arrow indicates upsampling, and the dashed arrow indicates the skip connection. Using the skip connection, features with the same resolution from the preceding node are combined. In addition, upsampling combines features from the preceding node with different resolutions. This multi-scale feature aggregation is an advance of UNet++.

**Figure 6 healthcare-11-00484-f006:**
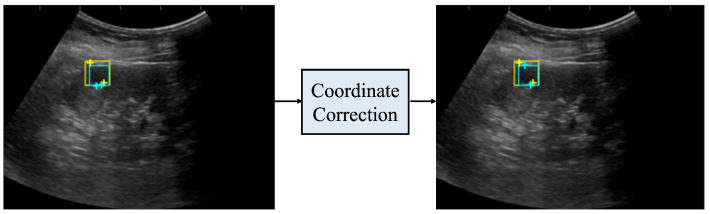
Coordinates before and after coordinate correction. The yellow rectangle is the true bounding box, the light blue rectangle is the bounding box predicted by the renal cyst detection model, the yellow crosses are the true salient landmark coordinates, and the light blue crosses are the salient landmark coordinates determined from the saliency map output by the salient landmark position prediction model. The left side of the figure shows the coordinates before correction and the right side shows the coordinates after correction. After correction, the distance between the two salient landmarks of the prediction is closer to the distance between the two salient landmarks of the true one.

**Figure 7 healthcare-11-00484-f007:**
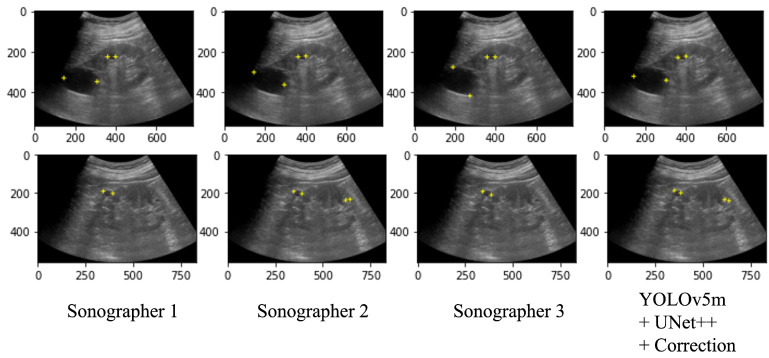
Ultrasound images with salient landmarks placed by sonographers and placed at coordinates predicted by the automated system.

**Figure 8 healthcare-11-00484-f008:**
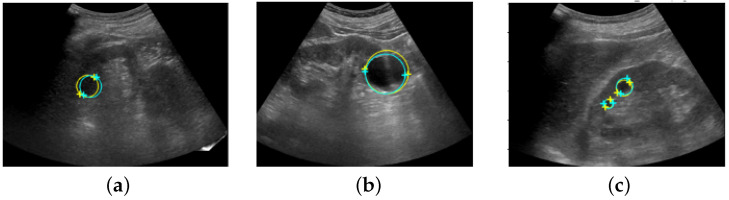
Ultrasound images with salient landmarks placed by sonographers and placed at coordinates predicted by the automated system. The yellow crosses are the coordinates of the salient landmarks placed by Sonographer 1 and the light blue crosses are the coordinates predicted by the automation system. Renal cysts are approximated by a circle with a diameter that is a straight line connecting the two points of salient landmarks. Yellow corresponds to placement by Sonographer 1 and light blue corresponds to prediction by the automated system. (**a**,**b**) show detection of a single renal cyst. (**c**) shows detection of two renal cysts.

**Table 1 healthcare-11-00484-t001:** Summary of related studies on deep learning-based analysis of ultrasound images.

Study	Object	Task
Ma et al. [3]	thyroid nodules	detection
Li et al. [4]	thyroid cancer	classification
Byra et al. [6]	breast cancer	classification
Liu et al. [7]	liver membrane	detection & classification
Chen et al. [14]	fetal lateral ventricles	measurement
Biswas et al. [15]	carotid	measurement
Leclerc et al. [16]	left ventricular	measurement
Jagtap et al. [17]	kidney	measurement
Akkasaligar et al. [18]	renal cysts	segmentation
Our study	renal cysts	landmark placement

**Table 2 healthcare-11-00484-t002:** The hyperparameter search and its results; the hyperparameters highlighted in bold were adopted.

Parameter	Range of Parameter to Be Searched
CNN architecture	VGG16, ResNet50, **densenet121**
Decode method	**transpose**, upsampling
Number of decoder filters	**(128, 64, 32, 16, 8)**, (256, 128, 64, 32, 16), (512, 256, 128, 64, 32)
Batch size	8 to 32 (**21**)

**Table 3 healthcare-11-00484-t003:** Results of the performance of the automated system in the test data (above) and performance comparison on the 100 selected images (below) between the automated system and the two radiologists (Sonographers 2 and 3) with the annotation of two specialists (Sonographer 1 and 4) as the ground truth. The position error and DLE were calculated only for the true positive predictions of salient landmark positions. * a statistical *t*-test examining the average of IoU values between the “YOLOv5m + UNet++ + Correction” combination method and other combinations, along with the calculated *p*-values.

	Detection Accuracy			Position Error [mm]		DLE [mm]	
	Precision	Recall	*p*-Value * (of *t*-Test for IoU Average)	Mean	Median	Mean	Median
YOLOv5s + UNet++	0.71	0.75	0.07	3.54 ± 2.81	2.54	1.22 ± 1.04	1.02
YOLOv5s + UNet++ + Correction	0.78	0.82	0.07	3.59 ± 2.81	2.63	1.19 ± 1.02	0.93
YOLOv5m + UNet++	0.81	0.81	0.26	3.15 ± 2.48	2.36	1.08 ± 0.83	0.90
YOLOv5m + UNet++ + Correction	0.85	0.86	-	3.22 ± 2.57	2.36	1.09 ± 0.80	0.89
YOLOv5l + UNet++	0.74	0.78	0.07	3.19 ± 2.57	2.51	1.06 ± 1.92	0.90
YOLOv5l + UNet++ + Correction	0.82	0.86	0.18	3.29 ± 2.66	2.60	1.13 ± 1.00	0.88
YOLOv5x + UNet++	0.70	0.73	0.004	3.05 ± 2.72	2.39	1.15 ± 0.87	1.05
YOLOv5x + UNet++ + Correction	0.77	0.81	0.04	3.24 ± 2.73	2.63	1.13 ± 0.89	0.91
		**Detection Accuracy**		**Position Error [mm]**		**DLE [mm]**	
		**Precision**	**Recall**	**Mean**	**Median**	**Mean**	**Median**
Sonographer 2		0.86	0.87	2.56 ± 2.76	1.42	1.21 ± 1.36	0.89
Sonographer 3		0.83	0.84	2.34 ± 2.63	1.53	0.95 ± 1.07	0.63
YOLOv5m + UNet++ + Correction		0.85	0.86	3.22 ± 2.57	2.36	1.09 ± 0.8	0.89

**Table 4 healthcare-11-00484-t004:** Results of post hoc manual evaluation by the most experienced radiologist.

	False	False	Corrected Coordinates		Position Error [mm]	
	Positive [Pair]	Negative [Pair]	Number of Images [Image]	Number of Landmarks [Point]	Mean	Median
Sonographer 2	3	3	16	18	9.09 ± 5.90	6.80
Sonographer 3	1	0	14	21	8.65 ± 5.40	7.24
YOLOv5m + UNet++ + Correction	1	1	16	22	8.49 ± 8.04	5.47

## Data Availability

Not applicable.

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
