# Peer review of "Automatic Detection and Measurement of Renal Cysts in Ultrasound Images: A Deep Learning Approach"

_healthcare, 2023, doi:10.3390/healthcare11040484_

Round 1
Reviewer 1 Report
This is an interesting research work, presenting a deep learning approach for automated assignment of salient landmarks on renal cysts. While the authors use a combination of already established methods, the novelty lies in applying it for the prediction of salient landmark positions for multiple renal cysts in ultrasound images without requiring prior segmentation. The manuscript can be considered for publication after addressing the below comments.
- The authors have stated that there are few studies in abdominal ultrasonography (line 80). Are there studies particularly for renal cysts? Please explicitly mention this. If yes, please also clarify the similarities and dissimilarities between the presented work and the existing works. This is an important point to be discussed in the introduction section itself.
- Referring to lines 103-105, please also mention the existing studies for salient landmark positions' prediction for multiple renal cysts even if their accuracy is lower than that from a radiologist. This will provide a complete literature review.
- Lines 157-158: How was the radius of the Gaussian distribution chosen?
- 2.6 coordinate correction: It is not clear if this methodology is a novel proposition by the authors. If yes, please explicitly state. If not, relevant work should be cited.
- What is meant by 'mAP' in line 335?
- Table 2: for any given model, why does the position error increase when correction is also incorporated?
Minor comments:
- Second paragraph of the introduction section could actually start with sentence at line 26: "studies have been conducted..."
- Line 35: Insert space between of and radiologists.
- Line 80: Mention relevant references for studies in abdominal ultrasonography.
- Line 176: The sentence needs to be corrected: 'whether' to be removed to convey the right message.
- Line 182: There should be 'small' before (YOLOv5s).
- Variables in equations (1)-(4) are not defined.
- Line 340: None of the hyperparameters in Table 1 are in bold. Please correct it.
- Figure 8: Please refer to the different images as (a), (b), (c).
Author Response
Dear Reviewers,
We greatly appreciate the reviewers’ constructive comments, which helped us to considerably improve the quality of our paper. The manuscript has been revised accordingly. The revisions in the main text are highlighted in red. The responses to all comments have been prepared and attached in the attachment. In the attachment, we present our point-by-point responses to each of the reviewers’ comments along with the details of all revisions made. The language has been revised by an English editing service.

Reviewer 2 Report
Authors have proposed a deep learning-based approach to detect and measure renal cysts in ultrasound images. I have the following comments that must be addressed in the revised paper.
Introduction
1. Line 60 and 65: Starting two adjacent paragraphs with several is inappropriate.
2. Pages 1 and 2: Authors have described many deep learning methods in various fields. How this survey relates to the problem?
3. Describes the background by illustrating the general area of research. It should also mention the importance of the selected research area by highlighting its critical factors.
4. Briefly overview the existing practices and limitations of the current practices in detecting renal cysts and measurement.
5. Explain the research gap.
6. How your proposed solution will solve the limitations highlighted?
7. Highlight the novelty of the proposed solution.
8. Lines 88-97: These lines should be added in the methodology part.
9. Line 103: This statement may need to be corrected. Please search google scholar and other databases, and you will find some papers addressing this issue. A review of their performance is required.
Materials and Methods
10. Line 148: Subsection 2.2 should be after subsection 2.3.
11. For equations 1 to 4, all variables must be adequately defined.
12. Line 156: defining parameters in the methodology section is inappropriate.
13. Line 200: How are all layers fine-tuned? What is the advantage of using the same grayscale image for all three RGB channels?
14. A brief description of Unet++ and its architecture is required.
15. Subsection 2.7 Dataset needs to be explained appropriately. Describe how the dataset is collected from the patients. How many images and how many cysts per image? Were there any images with no cysts? A demographical analysis of the patients is also required. Combine sections 2.7 and 2.13.
16. Subsection 2.7 while creating the ground truth, two human sonographers put the landmarks on an image. Since they are more than one, how to finalize the bounding box and landmarks from these sonographers? How to tackle human error and differences among the sonographers.
17. Line 350: What are the criteria for selecting a smaller number of images (2664 out of 170538 images)?
18. Line 354: Are images annotated by four sonographers or 18 radiation doctors and radiologists?
19. Line 360: There needs to be more clarity. Does the Testing dataset consists of 30% of 2664 images or only 100 images? Also, the process of annotation needs to be clarified.
Results
20. Line 303: The definition of AD needs to be clarified. Why is it needed when position error is already calculated? Instead of position error, relative position error should be used to know the error as a percentage. In table 2, we can only say that position error is low or high by knowing the actual values of landmark positions or lengths.
21. It may be beneficial for the readers if an analysis of the size of cysts against their detection accuracy is done.
22. Table 2, why are position errors high and AD values low? Errors must be reported as mean and standard deviation.
23. Ground truth is generated by two sonographers (1,4) and tested by two sonographers (1,3) in Table 2. How can we be sure that the detection of sonographers 1 and 4 is accurate, but the detection of sonographers 1 and 3 is not?
Author Response

(The authors gave the same response as above.)

Round 2
Reviewer 2 Report
The authors have answered most of my comments. However, a few comments are not appropriately addressed in the revised version.
1. Briefly overview the existing practices and limitations of the current practices in detecting renal cysts and measurement.
Response:
We believe the following statements that were described in the Materials and Methods section in the first version could address this and the next comments of Reviewer#2.
Further comment: A table that can provide a summary of existing techniques in detecting renal cysts should be included in the paper. Research gap means what these methods are not lacking, and the paper solves it
2. It is good to show the equations related to the deep learning framework with adequately explained parameters. Therefore, include the details in the revised manuscript.
3. YOLOv5 should be modified to handle one-channel grayscale images. In this way, the algorithm will be more optimized in terms of time and space complexity. Sending three same images to three channels is not professional.
Author Response
Dear Reviewers,
We appreciate the reviewer’s further comments. The manuscript has been revised accordingly. The revisions in the main text are highlighted in red. The responses to all comments have been attached in the attachment. In the attachment, we present our point-by-point responses to each of the reviewers’ comments along with the details of all revisions made. The language has been revised by an English editing service. The “Editing Certificate” is attached at the end.
